# VoxelTrack: Exploring Voxel Representation for 3D Point Cloud Object Tracking

Submission Id: 930

## ABSTRACT

Current LiDAR point cloud-based 3D single object tracking (SOT) methods typically rely on point-based representation network. Despite demonstrated success, such networks suffer from some fundamental problems: 1) It contains pooling operation to cope with inherently disordered point clouds, hindering the capture of 3D spatial information that is useful for tracking, a regression task. 2) The adopted set abstraction operation hardly handles density-inconsistent point clouds, also preventing 3D spatial information from being modeled. To solve these problems, we introduce a novel tracking framework, termed VoxelTrack. By voxelizing inherently disordered point clouds into 3D voxels and extracting their features via sparse convolution blocks, VoxelTrack effectively models precise and robust 3D spatial information, thereby guiding accurate position prediction for tracked objects. Moreover, VoxelTrack incorporates a dual-stream encoder with cross-iterative feature fusion module to further explore fine-grained 3D spatial information for tracking. Benefiting from accurate 3D spatial information being modeled, our VoxelTrack simplifies tracking pipeline with a single regression loss. Extensive experiments are conducted on three widely-adopted datasets including KITTI, NuScenes and Waymo Open Dataset. The experimental results confirm that VoxelTrack achieves state-of-the-art performance (88.3%, 71.4% and 63.6% mean precision on the three datasets, respectively), and outperforms the existing trackers with a real-time speed of 36 Fps on a single TITAN RTX GPU. The source code and model will be released.

## CCS CONCEPTS

• **Computing methodologies → Tracking**; **Vision for robotics**; **Hierarchical representations**;

## KEYWORDS

LiDAR point clouds, Single object tracking, Voxel representation, 3D spatial information

**ACM Reference Format:**
Anonymous Author(s). 2018. VoxelTrack: Exploring Voxel Representation for 3D Point Cloud Object Tracking. In *Proceedings of Make sure to enter the correct conference title from your rights confirmation emai (Conference acronym 'XX)*. ACM, New York, NY, USA, 10 pages. https://doi.org/XXXXXXX.XXXXXXX

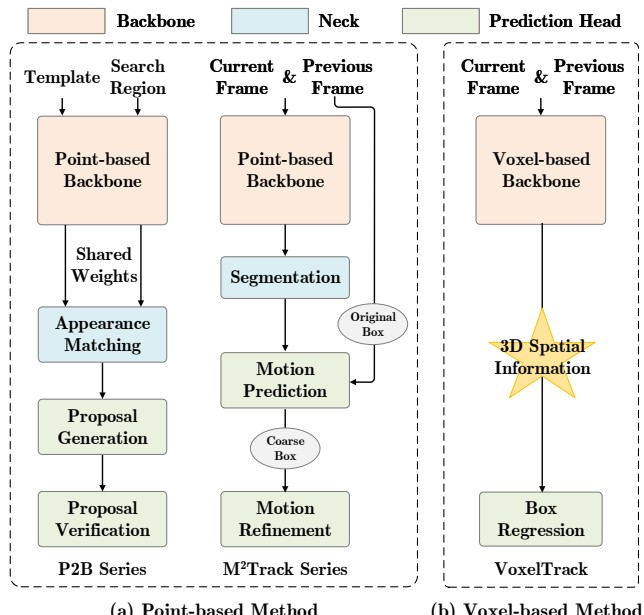

**Figure 1: Comparison between existing point-based tracking methods (a) and our proposed voxel-based tracking method (b). The point-based methods include P2B series and M²Track series. P2B series employs appearance matching techniques to generate proposals and verifies one as tracking result, while M²Track series models motion relation for tracking in a two-stage manner. In contrast, our VoxelTrack explores 3D spatial information through voxel-based representation for tracking, thereby simplifying the tracking pipeline with a single regression loss function.**

## 1 INTRODUCTION

Single object tracking (SOT) plays a pivotal role in various computer vision applications, such as autonomous driving [4, 59] and visual surveillance systems [46]. Early research works in SOT have predominantly focused on the 2D image domain [12, 60]. However, images are often disturbed by light and noise, making it difficult to track targets in the images. In recent years, with the rapid development of LiDAR sensors and considering that point cloud data is robust to light interference and environmental factors, many techniques [15, 19, 36, 41, 55, 56, 64] for 3D SOT have been proposed. Despite demonstrated success, these methods are built upon 2D SOT techniques, which may not be directly applicable to 3D SOT based on LiDAR point clouds, as point cloud data differs fundamentally from RGB image data. Therefore, it is crucial to develop

tracking techniques tailored to disordered and density-inconsistent point cloud data.

Currently, most of 3D SOT methods are based on point-based representation networks [34, 52, 55, 63], such as PointNet [39] and PointNet++ [40], which extract point features for subsequent task-oriented modules. As illustrated in the left of Fig. 1 (a), P2B [41] is an end-to-end tracking framework. It first encodes the semantic features of the point cloud in the template and search region by point representation based Siamese backbone [9], and then performs appearance matching of the template and the search region at the feature level. The proposals [16] are generated from the resulting search region enriched by template information, where a proposal is verified as the tracking result. Based on this strong framework, a series of trackers are presented, such as BAT [63], PTTR [65], GLT-T [36] that consists of point representation based Siamese backbone, appearance matching, proposals generation and verification. In contrast to P2B series, $M^2$Track [64] introduces a motion-centric paradigm, which explores motion cues instead of appearance matching for tracking. It predicts target motion from the concatenated point clouds of previous and current frames in a two-stage manner by using point features, as shown in the right of Fig. 1 (a).

In summary, previous research works have either used point features for appearance matching to guide tracking, or used point features to mine motion information. Although great success has been achieved in 3D SOT, point-based representation may be the sub-optimal for point cloud object tracking due to the following reasons: 1) Point representation-based networks rely on pooling operations to maintain the permutation invariance of disordered point clouds, thus encoding geometric structure information. However, the pooling operations tend to impair 3D spatial information of point clouds, which is essential for accurately regressing bounding boxes of point cloud objects. 2) The set abstraction operation employed in point presentation networks learns key point features through down-sampling, grouping and feed-forward blocks. Nonetheless, such set abstraction operation struggles to effectively handle the inherently density-inconsistent point clouds, thereby preventing 3D spatial information from being modeling.

To solve the above problems, we propose to leverage voxelized point clouds as input and employ voxel-based representation network for 3D SOT. We therefore introduce a novel tracking framework, termed VoxelTrack, which fully explores 3D spatial information of point clouds to guide target box regression for tracking, as shown in Fig. 1 (b). We first voxelize point clouds cropped from two consecutive frames and align them spatially, and then extract voxel features by a series of sparse convolution blocks. Leveraging the derived features incorporated by rich 3D spatial information of point clouds, we could perform box regression to predict target bounding box without any task-oriented module, such as proposal generation and verification, motion prediction and refinement. To further enhance 3D spatial information for accurate tracking, we incorporate dual-stream voxel representation learning network to explore fine-grained 3D spatial information. In addition, we perform layer-by-layer feature interaction for the two streams through a cross-iterative feature fusion module, enhancing the synchronization between dual-stream voxel features and thereby guiding a more accurate box regression.

We evaluate our proposed VoxelTrack on three widely-adopted datasets, including KITTI, NuScenes and Waymo Open Dataset. Experimental results demonstrate that VoxelTrack outperforms P2B/$M^2$Track by a significant margin of 28.0%/7.5%, 22.5%/9.8% mean success on the KITTI and NuScenes, respectively. Our method could also accurately track objects in complex point clouds scenes, such as those with sparse point clouds and distractors. Moreover, thanks to the removal of task-oriented modules, the proposed VoxelTrack runs at a real-time speed of 36 Fps on a single TITAN RTX GPU.

The main contributions of this work are summarized as follows:

- We propose VoxelTrack, a novel LiDAR point cloud tracking framework based on voxel representation. It simplifies tracking pipeline with a single regression loss function.
- We design a dual-stream encoder to extract multi-level voxel features in a cross-iterative fusion manner, capturing fine-grained 3D spatial information of point clouds to guide more accurate box regression.
- We conduct comprehensive experiments on KITTI, NuScenes and Waymo Open Dataset (WOD), demonstrating the superior performance of our proposed VoxelTrack.

## 2 RELATED WORK

### 2.1 2D Single Object Tracking

In recent years, research in the study of 2D object tracking [9, 12, 22, 50, 57, 60, 67] has become very mature. Most of the trackers [23, 24, 50, 57, 67] follow a Siamese-based network. SiamFC [1] as a pioneering work in the field, introduces a fully-convolution framework to achieve feature fusion and matching of template and search areas for object tracking. SiamRPN [25] first introduces a single-stage region proposal network (RPN) [16] to achieve object tracking by comparing the similarity between the current frame features and template features. By using a spatial-aware sampling strategy, SiamRPN++ [23] further improves the siamese-based network to remove the disturbance factors such as padding. Due to the extraordinary correlation modeling ability of Transformer [47] and the proposal of VIT [11], SimViT-Track [5] uses a similar method to feed templates and search areas into the ViT backbone to predict the location of targets. After that, many other notable variants [6, 13, 35, 49, 54] in 2D SOT also achieve the considerable success. However, it is non-trivial to apply these techniques to 3D SOT on LiDAR point clouds.

### 2.2 3D Single Object Tracking

Early 3D single object tracking (SOT) methods [2, 21, 29, 37] primarily operate within the RGB-D domain. While these approaches offer a promising research direction, RGB-D trackers may falter in tracking targets when the RGB-D information is compromised by factors such as lighting conditions or noise. With the advancements in LiDAR sensor technology, the point cloud domain has emerged as a pivotal domain for the evolution of 3D SOT, overcoming the aforementioned limitations. Pioneering this shift, SC3D [15] introduces the first Siamese tracker based solely on point clouds, matching feature distances between template and search regions while regularizing training through shape completion. Subsequently, P2B [41] introduces a 3D Region Proposal Network (RPN). Diverging from

SC3D, P2B adopts an end-to-end framework, employing a VoteNet [38] to generate a batch of high-quality candidate target boxes. Inspired by the success of P2B, numerous networks [18, 19, 32, 34, 36, 42, 48, 52, 63, 65] utilizing it as a baseline have been proposed. OSP2B [34] enhances P2B into a one-stage Siamese tracker by simultaneously generating 3D proposals and predicting center-ness scores. BAT [63] encodes distance information from point-to-box to augment correlation learning between template and search areas. PTTR [65], CMT [18], and STNet [20] explore different attention mechanisms to improve feature propagation. CXTrack [55] underscores the significance of contextual information on tracking performance, designing a target-centric transformer network to explore such information. MBPTrack [56] employs a memory mechanism for enhancing aggregation of previous spatial and temporal information. Despite the impressive performance achieved by these methods, they adhere to the Siamese paradigm, focusing either on designing intricate modules or incorporating additional point cloud features. However, owing to the sparse nature of point clouds, they may lack sufficient texture information for appearance matching. Fortunately, information pertaining to object motion is well retained. $M^2$Track introduces a motion-centric paradigm that explicitly models target motion between successive frames by directly fusing point clouds from two consecutive frames as input and utilizing PointNet [39] to predict relative target motion from the cropped target area. While these point representation-based tracking methods have demonstrated superior performance, they may still face challenges to deal with inherently disordered and sparsity-inconsistent point clouds. In this paper, we propose VoxelTrack, a novel voxel representation-based tracking framework.

## 2.3    Feature Representation on 3D Point Clouds

Presently, 3D perception approaches employing point cloud representation are primarily categorized into point-based methods and voxel-based methods. PointNet [39] and PointNet++[40] have paved the way for a multitude of vision tasks, including classification, segmentation, and detection, to be conducted directly on raw point cloud data. Building upon the success of these frameworks, a plethora of point-based representation methods have emerged[27, 31, 43, 45, 51, 53, 61, 62]. However, such point-based representation networks encounter inherent challenges. Given the sparsity and inconsistent density of point clouds, these methods heavily rely on point sampling and regional point aggregation operations, which are susceptible to time-consuming and computation-intensive point-wise feature duplication. On the other hand, voxel-based representation methods in 3D perception entail partitioning the input point cloud into a grid of uniformly sized voxels based on a predefined coordinate system, addressing the irregular data format issue. VoxelNet [66] lays the groundwork for target detection frameworks and serves as the backbone for numerous 3D detection methods. Sparse convolution techniques are employed in SECOND [58] to proficiently learn sparse voxel features from point clouds. Compared to point-based methods, voxel-based methods [8, 10, 30, 33] demand fewer memory and computing resources, rendering them potentially superior choices for point cloud object tracking task. To the best of our knowledge, VoxelTrack is the first attempt to utilize a pure voxel representation network for 3D SOT.

## 3    METHODOLOGY

In this section, we present our proposed VoxelTrack to implement the 3D SOT task. We first explicitly define the 3D SOT task in Sec. 3.1, followed by a review of two popular model series using point-based representation in Sec. 3.2. In Sec. 3.3, we describe our proposed framework VoxelTrack, consisting of three key components: Voxel Division, Multi-level Voxel Representation Learning and Box Regression. Finally, the implementation of the three components are detailed in Sec. 3.4, 3.5 and 3.6, respectively.

### 3.1    Problem Definition

The task of 3D SOT is defined: In a $T$-frame sequence of point clouds $\{\mathbf{P} \in \mathbb{R}^{N \times 3}\}_{t=1}^{T}$, an initial 3D bounding box $B_1 = (x_1, y_1, z_1, w_1, h_1, l_1, \theta_1)$ of a specific target is given in the first frame, tracking needs to accurately localize the target in subsequent frame and predict bounding boxes $\{B_t = (x_t, y_t, z_t, w_t, h_t, l_t, \theta_t)\}_{t=2}^{T}$ for the tracked target. The point cloud $\mathbf{P} \in \mathbb{R}^{N \times 3}$ is composed of $N$ points. $(x, y, z)$, $(w, h, l)$ and $\theta$ in bounding box represent the center coordinates, box size and angle. In the community, it is assumed that the target size remains constant across all frame. Therefore, only $(x, y, z)$ and $\theta$ are needed to predict for 3D bounding box.

### 3.2    Revisit Point Representation-based Trackers

**P2B series.** Existing P2B series trackers crop the point clouds in the first frame by using the given target box to get template region $P_1^{crop}$, and form search region $P_t^{crop}$ in current $t$-th frame by expanding target box predicted in previous $(t-1)$-th frame. The template and search region are then delivered into a point representation based Siamese network. Following, operations such as similarity calculations are typically employed to assess the appearance matching degree between $P_1^{crop}$ and $P_t^{crop}$. The proposals are generated from search region by offset learning. Finally, the proposal with the highest confidence is selected to generate tracking result. The whole process can be represented by the following equation:

$$F_{pro}(F_{match}(F_p(P_1), F_p(P_t))) \rightarrow (\Delta x_t, \Delta y_t, \Delta z_t, \Delta \theta_t) \quad (1)$$

where $F_p$, $F_{match}$ and $F_{pro}$ denote the point representation based backbone, appearance matching network and proposal network. The output $(\Delta x_t, \Delta y_t, \Delta z_t, \Delta \theta_t)$ is added to $(x_{t-1}, y_{t-1}, z_{t-1}, \theta_{t-1})$ to yield tracking box $(x_t, y_t, z_t, \theta_t)$ in current frame.

**$M^2$Track series.** Current $M^2$Track do not crop the template region, it concatenates $P_{t-1}^{crop}$ and $P_t^{crop}$ into one input $P_{t-1,t}$. This approach distinguishes between previous frame target, current frame target and background points via joint spatial-temporal learning. Afterwards, they leverage a two-stage network to explore motion cue instead of appearance matching for tracking. A coarse box offset $(\Delta x_t, \Delta y_t, \Delta z_t, \Delta \theta_t)_{coarse}$ and a fine box offset $(\Delta x_t, \Delta y_t, \Delta z_t, \Delta \theta_t)_{fine}$ are predicted from the two stages, respectively. The whole process can be represented by the following equation:

$$F_{motion}(F_{seg}(F_p(P_{t-1,t}))) \rightarrow (\Delta x_t, \Delta y_t, \Delta z_t, \Delta \theta_t) \quad (2)$$

where $F_{seg}$ and $F_{motion}$ denote the segmentation network and motion inference network.

**Figure 2: Overall of our proposed voxel representation based tracking framework VoxelTrack. It consists of voxel division, multi-level voxel feature learning and box regression components. "CIF" denotes cross-iterative feature fusion module, where the last one performs single-direction fusion from small voxel (high resolution) branch to large voxel (low resolution) branch.**

### 3.3 Proposed VoxelTrack

Different from the above two series of trackers, our proposed VoxelTrack leverages voxel representation for point clouds to perform tracking and simplify the tracking framework into:

$$F_{reg}(F_v(P_{t-1,t})) \rightarrow (\Delta x_t, \Delta y_t, \Delta z_t, \Delta \theta_t) \quad (3)$$

where $F_v$ and $F_{reg}$ denote the voxel representation backbone and box regression network. The overall architecture of our proposed framework is illustrated in Fig. 2. It consists of three key components: voxel division, multi-level voxel representation learning and box regression. More concretely, we first divide point clouds in frame $t-1$ and $t$ into multi-level voxels and concatenate voxels in each level. Then, we propose a multi-level voxel representation learning module to capture 3D spatial information dependencies for tracked objects. Finally, we directly regress the target box, *i.e,*, predict $(\Delta x_t, \Delta y_t, \Delta z_t, \Delta \theta_t)$ by using a single regression loss.

### 3.4 Voxel Division.

The previous frame point cloud $P_{t-1}$ and current frame point clouds $P_t$ are first divided into voxels $\{\mathbf{V}_{t-1}, \mathbf{V}_t\}$ with spatial resolution of $W \times L \times H$:

$$(i, j, k) = (\lfloor \frac{x}{\Delta_W} \rfloor, \lfloor \frac{y}{\Delta_L} \rfloor, \lfloor \frac{z}{\Delta_H} \rfloor) \quad (4)$$

Where $\lfloor \cdot \rfloor$ denotes the floor function, (i, j, k) is the voxelized coordinate, (x,y,z) is the original coordinate of the point, $\Delta$ is the voxel size. Then we concatenate the previous frame $\mathbf{V}_{t-1} \in \mathbb{R}^{W \times L \times H \times 3}$ and the current frame $\mathbf{V}_t \in \mathbb{R}^{W \times L \times H \times 3}$ along the channel dimension as input $\mathbf{V}_{t-1,t} \in \mathbb{R}^{W \times L \times H \times 6}$. Considering that the point clouds in each frame may be relatively sparse, especially in distant scenes, we find that the previous pooling and set abstraction operations (involving farthest point sampling and feature forward) in point-based methods should be improved, preventing 3D spatial information from being modeled. We therefore introduce a simple dynamic voxel feature encoder to tackle non-empty voxels after voxelization. In each voxel, it averages the values of all points and

then reassigns the average value to each point. As a result, the number of points in each voxel is dynamic and will not be reduced:

$$\underset{j \in R^6}{Channel}(\mathbf{p}_i) = \frac{1}{N} \sum_{k=1}^{N} \underset{j \in R^6}{Channel}(\mathbf{p}_k) \quad (5)$$

By leveraging the dynamic voxel feature encoder, all points and spatial information are well retained without introducing information loss, so that VoxelTrack can learn rich features with 3D spatial information and guide accurate box regression for tracking.

### 3.5 Dual-Stream Voxel Representation Learning

**Dual-stream Encoder.** To extract rich spatial information from voxelized point clouds by making full use of their ordered spatial structure, we design a dual-stream encoder specifically for the modeling of fine-grained 3D spatial information. As shown in Fig. 3, we first divide the point clouds into two scales $\mathbf{V}^{large}$ and $\mathbf{V}^{small}$. After that, we generate two inputs $\mathbf{V}_{t-1,t}^{large} \in \mathbb{R}^{W_l \times L_l \times H_l \times 6}$ and $\mathbf{V}_{t-1,t}^{small} \in \mathbb{R}^{W_s \times L_s \times H_s \times 6}$ for the dual-stream encoder. Due to the sparsity of the point cloud, a lot voxels are empty. We then employ slightly-modified VoxelNext [7] as feature extraction backbone, which uses 3D sparse convolution layers [17] instead of 3D convolution layers to encode voxel features. According to our design, two different scales of voxel inputs are extracted features by two similar backbones, respectively to focus on learning multi-scale features $\{\mathbf{F}^{large}, \mathbf{F}^{small}\}_{s=1}^N$. Specifically, our backbone is consisted of $N$ stages, where the spatial information $\mathbf{F}^L$ is down-sampled to half after each stage. Meanwhile, the number of channels is doubled to enhance the voxel feature representation ability. The scale transformation for $\mathbf{F}_s^{large}$ can be formulated as:

$$\mathbf{F}_{s+1}^{large} \in \mathbb{R}^{\frac{W}{2} \times \frac{L}{2} \times \frac{H}{2} \times 2C} = \mathbf{SpConv}(\mathbf{F}_s^{large} \in \mathbb{R}^{W \times L \times H \times C}) \quad (6)$$

where $s \in \{0, 1, 2\}$, and the scale transformation for $\mathbf{F}_s^{small}$ is similar to Eq. 6.

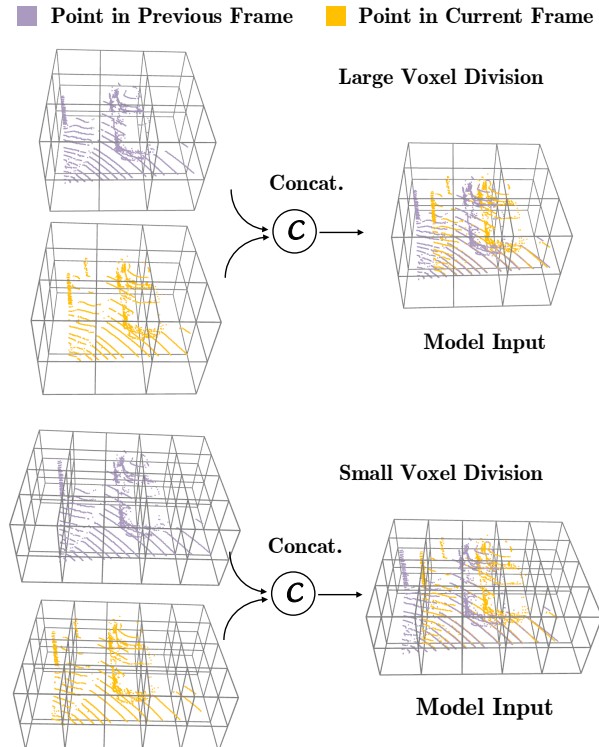

Figure 3: Illustration of large voxel and small voxel based inputs for dual-stream encoder. The inputs are denoted by $\mathbf{V}_{t-1,t}^{large} \in \mathbb{R}^{W_l \times L_l \times H_l \times 6}$ and $\mathbf{V}_{t-1,t}^{small} \in \mathbb{R}^{W_s \times L_s \times H_s \times 6}$, respectively.

**Cross-iterative Feature Fusion.** Although the utilization of two independent branches enables more comprehensive extraction of various features, the lack of interdependence between these branches impedes the effective aggregation of features. As one of the most successful feature aggregation module, the feature pyramid network (FPN) [28] extracts multi-scale semantic information from different layers of feature through both top-down and bottom-up feature propagation mechanisms. In contrast, our VoxelTrack aims at interacting features of the dual streams to enhance the representation of 3D spatial information for the tracking task, rather than fusing features of different stages. Therefore, we develop a cross-iterative feature fusion (CIF) module, which can iteratively fuse the features of each stage to enhance the synchronization of dual-stream features.

As show in Fig. 4, after being encoded by sparse convolution block of $s$-th stage, the voxel features $\mathbf{F}_s^{large}$ and $\mathbf{F}_s^{small}$ are formed in the two branches, respectively. To fuse $\mathbf{F}_s^{large}$ and $\mathbf{F}_s^{small}$, we first align the scale of two feature map groups. For the $\mathbf{F}_s^{small}$ with high-resolution, it is down-sampled to the size of $\mathbf{F}_s^{large}$ by 3D Average Pooling. Similarly, we up-sample the $\mathbf{F}_s^{large}$ with low-resolution by linear interpolation operation. Compared to using the convolution layers for sampling, our method is able to better preserve features of different scale and reduce the consumption of computational resources. Detailed experiments and analysis can refer to Tab. 6).

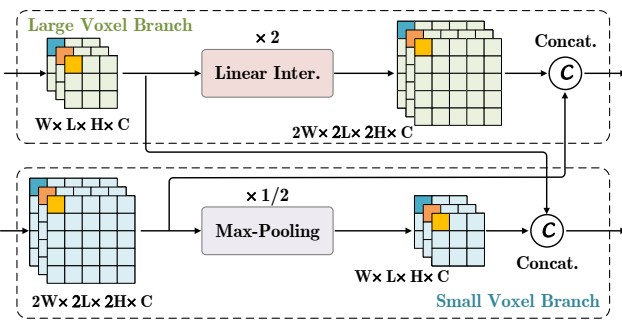

Figure 4: Illustration of cross-iterative feature fusion. It utilizes a pooling operation to down-sample the large-scale 3D feature maps within the small voxel branch, which are then concatenated with the small-scale 3D feature maps of the large voxel branch. Correspondingly, a linear interpolation operation is employed to fuse feature from the large voxel branch to the small voxel branch.

It is important to note that we use only the down-sampling in the last stage to obtain the output features of the backbone. To make a better fusion, we then employ convolution layer to further process features.

## 3.6 Box Regression

With the obtained encoded features, we convert 3D voxel features to 2D BEV features $\mathbf{F}^{BEV}$. Thanks to 3D spatial information being modeled, we directly predict the target position $(\Delta x, \Delta y, \Delta z, \Delta \theta)$ in a one-stage manner. Specifically, we first aggregate the spatial information by global max pooling. Then we apply a MLP to derive the target position. Finally, we leverage a residual log-likelihood estimation [26] function $\mathcal{F}$ to calculate training loss $\mathcal{L}$:

$$\mathcal{L} = \mathcal{F}_{rle}((\Delta x_t, \Delta y_t, \Delta z_t, \Delta \theta_t), (\hat{\Delta x}_t, \hat{\Delta y}_t, \hat{\Delta z}_t, \hat{\Delta \theta}_t)) \quad (7)$$

where $(\Delta x_t, \Delta y_t, \Delta z_t, \Delta \theta_t)$ and $(\hat{\Delta x}_t, \hat{\Delta y}_t, \hat{\Delta z}_t, \hat{\Delta \theta}_t)$ denote the prediction parameters and label, respectively.

## 4 EXPERIMENT

## 4.1 Experimental setting

**Implementation Details.** For model inputs, we first crop the point cloud $P_{t-1}$ and $P_t$ with a range of [(4.8,-4.8),(4.8,-4.8),(1.5,-1.5)] for Car category and [(1.92,-1.92),(1.92,-1.92),(1.5,-1.5)] for Pedestrian category. We then set $W_l$, $L_l$ and $H_l$ to 64, 64 and 10 for large voxel branch, and set $W_s$, $L_s$ and $H_s$ to 128, 128 and 20 for small voxel branch. The dual-stream encoder is implemented by a series of sparse convolution blocks [66]. We train our VoxelTrack using a AdamW optimizer on eight TITAN RTX GPUs, with a batch size of 128. The initial learning rate is set to 1e-4, which is reduced by a factor of 5 every 40 epochs. More implementation details can be found in appendix.

**Datasets.** To evaluate the performance of our tracking network, we conducted comprehensive and detailed experiments and analyses on three large-scale and widely used datasets, including KITTI [14] , NuScenes [3] and Waymo Open Dataset (WOD) [44]. KITTI consists of 21 training scene sequences and 29 test scenes sequences.

Table 1: Comparisons with state-of-the-art methods on KITTI dataset [14]. Red and blue denote the best performance and the second-best performance, respectively. Success / Precision are used for evaluation.

| Tracker | Source | Car [6,424] | Pedestrian [6,088] | Van [1,248] | Cyclist [308] | Mean [14,068] | Hardware | Fps |
|---|---|---|---|---|---|---|---|---|
| SC3D [15] | CVPR'19 | 41.3 / 57.9 | 18.2 / 37.8 | 40.4 / 47.0 | 41.5 / 70.4 | 31.2 / 48.5 | GTX 1080Ti | 2 |
| P2B [41] | CVPR'20 | 56.2 / 72.8 | 28.7 / 49.6 | 40.8 / 48.4 | 32.1 / 44.7 | 42.4 / 60.0 | GTX 1080Ti | 40 |
| MLVSNet [52] | ICCV'21 | 56.0 / 74.0 | 34.1 / 61.1 | 52.0 / 61.4 | 34.4 / 44.5 | 45.7 / 66.6 | GTX 1080Ti | 70 |
| BAT [63] | ICCV'21 | 60.5 / 77.7 | 42.1 / 70.1 | 52.4 / 67.0 | 33.7 / 45.4 | 51.2 / 72.8 | RTX 2080 | 57 |
| PTTR [65] | CVPR'22 | 65.2 / 77.4 | 50.9 / 81.6 | 52.5 / 61.8 | 65.1 / 90.5 | 57.9 / 78.2 | Tesla V100 | 50 |
| V2B [19] | NeurIPS'21 | 70.5 / 81.3 | 48.3 / 73.5 | 50.1 / 58.0 | 40.8 / 49.7 | 58.4 / 75.2 | TITAN RTX | 37 |
| CMT [18] | ECCV'22 | 70.5 / 81.9 | 49.1 / 75.5 | 54.1 / 64.1 | 55.1 / 82.4 | 59.4 / 77.6 | GTX 1080Ti | 32 |
| GLT-T [36] | AAAI'23 | 68.2 / 82.1 | 52.4 / 78.8 | 52.6 / 62.9 | 68.9 / 92.1 | 60.1 / 79.3 | GTX 1080Ti | 30 |
| OSP2B [34] | IJCAI'23 | 67.5 / 82.3 | 53.6 / 85.1 | 56.3 / 66.2 | 65.6 / 90.5 | 60.5 / 82.3 | GTX 1080Ti | 34 |
| STNet [20] | ECCV'22 | 72.1 / 84.0 | 49.9 / 77.2 | 58.0 / 70.6 | 73.5 / 93.7 | 61.3 / 80.1 | TITAN RTX | 35 |
| M$^2$Track [64] | CVPR'22 | 65.5 / 80.8 | 61.5 / 88.2 | 53.8 / 70.7 | 73.2 / 93.5 | 62.9 / 83.4 | Tesla V100 | 57 |
| SyncTrack [32] | ICCV'23 | 73.3 / 85.0 | 54.7 / 80.5 | 60.3 / 70.0 | 73.1 / 93.8 | 64.1 / 81.9 | TITAN RTX | 45 |
| CorpNet [48] | CVPRw'23 | 73.6 / 84.1 | 55.6 / 82.4 | 58.7 / 66.5 | 74.3 / 94.2 | 64.5 / 82.0 | TITAN RTX | 36 |
| CXTrack [55] | CVPR'23 | 69.1 / 81.6 | 67.0 / 91.5 | 60.0 / 71.8 | 74.2 / 94.3 | 67.5 / 85.3 | RTX 3090 | 29 |
| **VoxelTrack** | Ours | 72.5 / 84.7 | 67.8 / 92.6 | 69.8 / 83.6 | 75.1 / 94.7 | 70.4 / 88.3 | TITAN RTX | 36 |
| **Improvement** | | ↓0.8 / ↓0.3 | ↑0.8 / ↑1.1 | ↑9.5 / ↑11.8 | ↑0.8 / ↑0.4 | ↑2.9 / ↑3.0 | | |

Table 2: Comparisons with state-of-the-art methods on Waymo Open Dataset [44].

| Tracker | Vehicle | | | | Pedestrian | | | | Mean [427,483] |
|---|---|---|---|---|---|---|---|---|---|
| | Easy [67,832] | Medium [61,252] | Hard [56,647] | Mean [185,731] | Easy [85,280] | Medium [82,253] | Hard [74,219] | Mean [241,752] | |
| P2B [41] | 57.1 / 65.4 | 52.0 / 60.7 | 47.9 / 58.5 | 52.6 / 61.7 | 18.1 / 30.8 | 17.8 / 30.0 | 17.7 / 29.3 | 17.9 / 30.1 | 33.0 / 43.8 |
| BAT [63] | 61.0 / 68.3 | 53.3 / 60.9 | 48.9 / 57.8 | 54.7 / 62.7 | 19.3 / 32.6 | 17.8 / 29.8 | 17.2 / 28.3 | 18.2 / 30.3 | 34.1 / 44.4 |
| V2B [19] | 64.5 / 71.5 | 55.1 / 63.2 | 52.0 / 62.0 | 57.6 / 65.9 | 27.9 / 43.9 | 22.5 / 36.2 | 20.1 / 33.1 | 23.7 / 37.9 | 38.4 / 50.1 |
| STNet [20] | 65.9 / 72.7 | 57.5 / 66.0 | 54.6 / 64.7 | 59.7 / 68.0 | 29.2 / 45.3 | 24.7 / 38.2 | 22.2 / 35.8 | 25.5 / 39.9 | 40.4 / 52.1 |
| CXTrack [55] | 63.9 / 71.1 | 54.2 / 62.7 | 52.1 / 63.7 | 57.1 / 66.1 | 35.4 / 55.3 | 29.7 / 47.9 | 26.3 / 44.4 | 30.7 / 49.4 | 42.2 / 56.7 |
| **VoxelTrack** | 65.4 / 72.9 | 57.6 / 66.2 | 56.2 / 66.9 | 60.0 / 69.1 | 44.2 / 66.5 | 36.2 / 57.0 | 32.5 / 53.4 | 37.9 / 59.3 | 47.5 / 63.6 |
| **Improvement** | ↓ 0.5 / ↑0.2 | ↑0.1 / ↑0.2 | ↑1.6 / ↑2.2 | ↑0.3 / ↑0.9 | ↑8.8 / ↑11.2 | ↑6.5 / ↑9.1 | ↑6.2 / ↑9.0 | ↑7.2 / ↑9.9 | ↑5.3 / ↑6.9 |

Following the previous works, we divide the training sequence into three subsets, sequences 0-16 for training, 17-18 for validation, and 19-20 for testing. Compared to KITTI, the other two datasets are larger and contain more challenging scenes. For the NuScenes, it contains 700/150 scenes for training / testing. For the WOD, it includes 1121 tracklets that are classified into easy, medium, and hard subsets based on the sparsity of point clouds. These configurations adhere to established methods to maintain a fair comparison.

**Metrics.** We use one pass evaluation (OPE) as an approach to evaluate the performance of our model, simultaneously use of both Success and Precision metrics. Success is calculated by the intersection over union (IOU) between the ground truth bonding box and the predicted bounding box. Precision is calculated as the distance between the centers of the two bounding boxes.

## 4.2 Comparison with State-of-the-arts

**Results on KITTI.** We compare the proposed VoxelTrack with existing point representation-based state-of-the-art methods, and present a comprehensive analysis towards the performance of these methods on all categories, including Car, Pedestrian, Van and Cyclist. As show in Tab. 1, our VoxelTrack demonstrates predominant performance across various categories, achieving a mean Success rate of 70.4% and a mean Precision rate of 88.3% in the KITTI dataset, respectively. This is due to that voxel-based representation captures accurate 3D spatial information, which is more suited for disordered and density-inconsistent point clouds. Compared to point-based representation based methods P2B [41] and M$^2$Track [64], VoxelTrack achieves 28.0% and 7.5% performance gains in terms of mean Success, while running at a real-time speed. In addition, our method exhibits significant performance advantage compared to recent P2B series work CXTrack [55]. Note that, VoxelTrack obviously outperforms the previous best method by in the Van category, which implies that our method can achieve good performance without the large dataset training.

**Results on WOD.** To demonstrate the applicability of our proposed VoxelTrack method, we evaluate the KITTI trained Car and

**Table 3: Comparisons with state-of-the-art methods on NuScenes dataset [3].**

| Tracker | Car [64,159] | Pedestrian [33,227] | Truck [13,587] | Trailer [3,352] | Bus [2,953] | Mean [117,278] |
|---|---|---|---|---|---|---|
| SC3D [15] | 22.3 / 21.9 | 11.3 / 12.6 | 30.6 / 27.7 | 35.3 / 28.1 | 29.3 / 24.1 | 20.7 / 20.2 |
| P2B [41] | 38.8 / 43.2 | 28.4 / 52.2 | 43.0 / 41.6 | 49.0 / 40.0 | 32.9 / 27.4 | 36.5 / 45.1 |
| PTT [42] | 41.2 / 45.2 | 19.3 / 32.0 | 50.2 / 48.6 | 51.7 / 46.5 | 39.4 / 36.7 | 36.3 / 41.7 |
| BAT [63] | 40.7 / 43.3 | 28.8 / 53.3 | 45.3 / 42.6 | 52.6 / 44.9 | 35.4 / 28.0 | 38.1 / 45.7 |
| GLT-T [36] | 48.5 / 54.3 | 31.7 / 56.5 | 52.7 / 51.4 | 57.6 / 52.0 | 44.5 / 40.1 | 44.4 / 54.3 |
| PTTR [65] | 51.9 / 58.6 | 29.9 / 45.1 | 45.3 / 44.7 | 45.9 / 38.3 | 43.1 / 37.7 | 44.5 / 52.1 |
| $M^2$Track [64] | 55.8 / 65.1 | 32.1 / 60.9 | 57.4 / 59.5 | 57.6 / 58.2 | 51.4 / 51.4 | 49.2 / 62.7 |
| **VoxelTrack** | 63.9 / 71.6 | 46.8 / 75.9 | 64.8 / 65.9 | 69.5 / 64.3 | 60.1 / 57.7 | 59.0 / 71.4 |
| **Improvement** | ↑8.1 / ↑6.5 | ↑14.7 / ↑15.0 | ↑7.4 / ↑6.4 | ↑11.9 / ↑6.1 | ↑8.7 / ↑6.3 | ↑9.8 / ↑8.7 |

**Table 4: Ablation of dual-stream voxel representation. "Single" means that only a small voxel branch is used, while "Dual" denotes the use of dual-stream encode under different ratios for large and small voxel branches.**

| Branch | Ratio | Car | Pedestrian | Van | Cyclist |
|---|---|---|---|---|---|
| Single | - | 69.1 / 80.2 | 63.5 / 88.2 | 65.4 / 76.7 | 72.5 / 90.1 |
| Dual | 1.5 | 71.8 / 83.5 | 67.3 / 91.7 | 68.1 / 80.2 | 73.1 / 91.4 |
| | 2.0 | **72.5 / 84.7** | **67.8 / 92.6** | **69.8 / 83.6** | **75.1 / 94.7** |
| | 2.5 | 70.9 / 82.6 | 65.4 / 90.1 | 68.7 / 81.9 | 74.2 / 93.6 |

**Table 5: Ablation of CIF module. "$S_n$" denotes $n$-th stage.**

| $S_1$ | $S_2$ | $S_3$ | Car | Pedestrian | Van | Cyclist |
|---|---|---|---|---|---|---|
| | | ✓ | 70.5 / 81.8 | 57.6 / 84.9 | 66.2 / 79.4 | 72.1 / 91.2 |
| | ✓ | ✓ | 70.1 / 81.6 | 65.0 / 88.7 | 68.1 / 81.6 | 74.5 / 93.6 |
| ✓ | ✓ | ✓ | **72.5 / 84.7** | **67.8 / 92.6** | **69.8 / 83.6** | **75.1 / 94.7** |

**Table 6: Ablation of variant design for CIF module. "Left+Right" represents that "Left" operation is used for up-sampling and "Right" operation is used for down-sampling, respectively. "UpConv" and "Lerp" denote transpose convolution and linear interpolation.**

| Variant | Car | Pedestrian | Van | Cyclist |
|---|---|---|---|---|
| UpConv + Conv | 71.1 / 82.8 | 65.6 / 90.2 | 68.5 / 81.4 | 74.8 / 94.0 |
| UpConv + Pool | 69.4 / 81.7 | 66.1 / 91.2 | 67.1 / 80.2 | 74.6 / 93.2 |
| Lerp + Conv | 68.8 / 80.3 | 66.7 / 92.0 | 68.1 / 81.0 | 74.3 / 93.7 |
| Lerp + Pool | **72.5 / 84.7** | **67.8 / 92.6** | **69.8 / 83.6** | **75.1 / 94.7** |

leading performance in all categories. The results of this experiment demonstrate that our method can accurately and robustly track objects even in complex scenes.

## 4.3 Exploration Studies

**Effectiveness of Dual-Stream Voxel Representation.** The proposed VoxelTrack leverages voxel-based representation to model spatial information and achieves direct regression of target box. As reported in Tab. 4, even with single-branch voxel encoding, VoxelTrack still presents favorable performance, such as 69.1% and 80.2% values in terms of Success and Precision. When using dual-stream encoder, performance is further improved. Here, we ablate the ratio between large and small voxel branches. In fact, both too large and too small ratios cause some degree of interference in the synchronization between the two-stream features. According to Tab. 4, when ratio is set 2, VoxelTrack achieves the best performance on four categories. Therefore, we set ratio to 2 for all experiments if not specified.

**Analysis of CIF Module.** To further analyze the influence of the cross-iterative feature fusion (CIF) module on our proposed VoxelTrack, we conduct an ablation study on the KITTI dataset. As reported in Tab. 5, VoxelTrack achieves best performance with iterative interaction fusion in each stage. This implies that CIF can effectively enhance the synchronization between the dual-stream features, thereby exploring fine-grained 3D spatial information for tracking. When only interacting with the dual-stream features in the last stage, the performance across different categories is reduced, notably on the Pedestrian category, by 10.2 and 7.7% in terms of Success and Precision, respectively. We consider that

Pedestrian model on WOD, following common practice in the community [19, 55]. We select some representative methods for comparison, including CXTrack [55], STNet [20], V2B [19], BAT [63] and P2B [41]. The experiment results are shown in the Tab. 2. Our VoxelTrack achieves best performance with a mean Success and Precision of 47.5% and 63.6%. Compared to existing point representation base methods, VoxelTrack presents performance improvements across vehicle and pedestrian categories with varying degrees of complexity. This is attributed to the higher generalization of voxel representation compared to point representation to unseen scenes, proving the potential of the proposed voxel representation-based tracking framework.

**Results on NuScenes.** We further explore the various capabilities of VoxelTrack on the NuScenes dataset. Because NuScenes contains a large number of complex and diverse scenes, it becomes a more challenging dataset for 3D SOT. We choose the state-of-the-art trackers that have reported performance on this dataset as comparisons: SC3D [15], P2B [41], PTT [42], BAT [63], GLT-T [36], PTTR [65] and $M^2$Track [64]. As show in Tab. 3, our VoxelTrack demonstrates great performance with the mean Success and Precision rates of 59.06% and 71.39%. Notably, VoxelTrack exhibits the

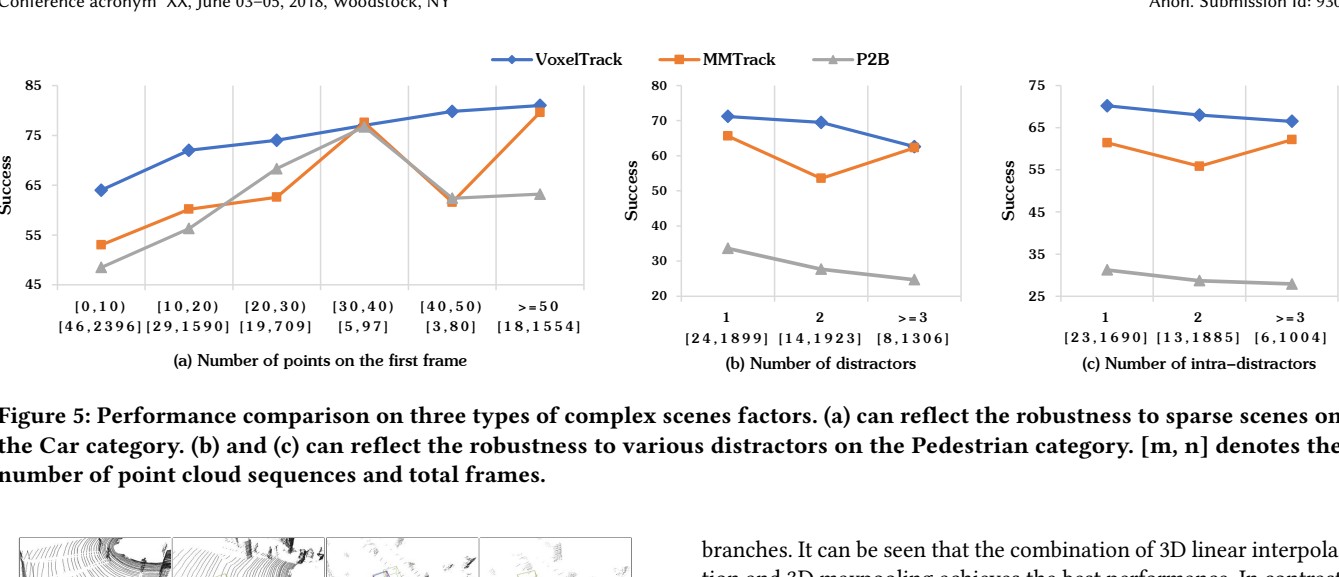

**Figure 5: Performance comparison on three types of complex scenes factors. (a) can reflect the robustness to sparse scenes on the Car category. (b) and (c) can reflect the robustness to various distractors on the Pedestrian category. [m, n] denotes the number of point cloud sequences and total frames.**

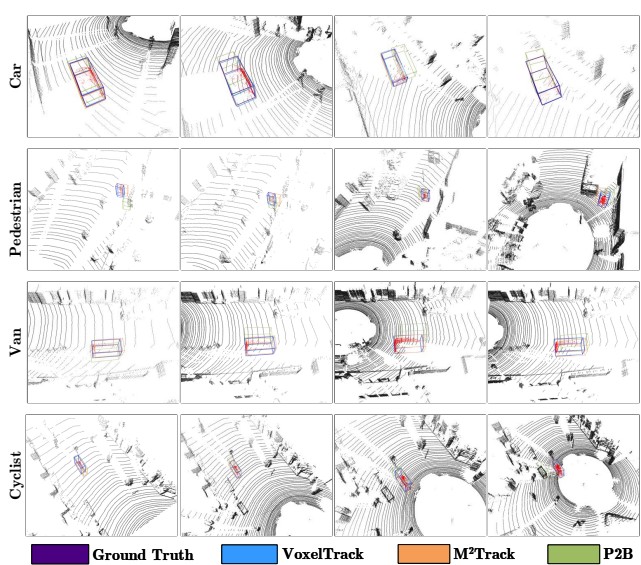

**Figure 6: Tracking visualization comparison across four categories on the KITTI dataset. The bounding box of VoxelTrack fits the ground truth box better than comparative methods.**

when the dual-stream features in previous stages are not fused, the high-resolution spatial information is distracted, which can lead to less accurate predictions of yaw angle, thus affecting Success metric more compared to Precision metric. The reason for the small impact on Success metric for the Car category is that the yaw angle of the cars does not change significantly in a point cloud sequence and is relatively easy to predict.

**Variant Design of CIF Module.** In our CIF module, the sampling method will affect the synchronization of feature interaction. As show in Tab. 6, we ablate two commonly used up-sampling methods and down-sampling methods, respectively. We choose 3D transposed convolution and 3D linear interpolation as alternatives for up-sampling methods, 3D convolution and 3D maxpooling for down-sampling methods. We generate four candidate variants by permuting and combining them. These variants will be applied to the CIF module fuse the feature maps extracted by the dual-stream

branches. It can be seen that the combination of 3D linear interpolation and 3D maxpooling achieves the best performance. In contrast to the convolutional approach, these two methods maintain feature semantic space, which facilitates the collaborative modeling of 3D spatial information required for tracking between the dual-stream features.

**Robustness to Sparsity and Distractors.** Considering that most point clouds in real scenes are sparse and contained with distractors, generally testing the performance on the test dataset may lack reliability in practical applications. Therefore, it is necessary to analyze the robustness of model to sparse point clouds and distractors. Following [41], we divide the Car category dataset into six sparsity levels, while divide the Pedestrian category dataset into three inter-class and extra-class distractors levels. As show in Fig. 6, we compare our VoxelTrack with $M^2$Track [64] and P2B [41]. VoxelTrack performs better in complex scenes (a), specially for extremely sparse scenes with fewer than 20 points. For (b) and (c), our method exhibits consistent performance advantage regardless of how many interfering objects the scene contains. These all results demonstrate the potential of the proposed method for practical applications.

**Visualization Analysis.** In Fig. 5, we show visualization results on the KITTI dataset. We obtain LiDAR point cloud sequences from each category, and then compare the ground truth box with three prediction bounding boxes of P2B, $M^2$Track and VoxelTrack. Our VoxelTrack can more accurately and robustly track objects across all categories than comparative methods, intuitively demonstrating the effectiveness of our proposed framework.

## 5 CONCLUSION

This paper presents a novel voxel representation based tracking framework, termed VoxelTrack. The novel framework leverages voxel representation to explore 3D spatial information to guide direct box regression for tracking. Moreover, It incorporates a dual-stream encoder with a cross-iterative feature fusion module to further model fine-grained 3D spatial information. Through extensive experiments and analyses, we prove that our proposed VoxelTrack effectively handles disordered and density-inconsistent point clouds, thereby exhibiting the state-of-the-art performance on three published datasets and significantly outperforming the previous point representation based methods.

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

Received 20 February 2007; revised 12 March 2009; accepted 5 June 2009

