# OpenReview forum: "VoxelTrack: Exploring Multi-level Voxel Representation for 3D Point Cloud Object Tracking"
_acmmm.org/ACMMM/2024/Conference — MM2024 Poster_

### Official Review · Reviewer_GcSb · 2024-05-23

**Rating:** 4
**Confidence:** 3

**Summary:**

Compared to the mainstream point-based 3D point cloud tracking, this paper proposed a novel voxel-based framework to capture rich spatial information. Besides, a dual-stream encoder is proposed to extract multi-level voxel features in a cross-iterative fusion manner. Extensive experiments are conducted on three datasets to demonstrate the effectiveness of the proposed VoxelTrack.

**Strengths:**

1. Using a voxel-based backbone to extract features is reasonable and has been demonstrated to be more effective than a point-based backbone.
2. The paper is well-written and easy to understand.
3. The experiments are well conducted and reasonable to support the goals of the proposed components.
4. The dual-stream encoder enhances the multi-scale information and boosts the performance.

**Limitations:**

1. It is suggested to replace the backbone of point-based methods with a voxel-based backbone or replace the backbone of VoxelTrack with the point-based backbone to further demonstrate the effectiveness of the proposed voxel representation.
2. It is better to provide more discussion about the motivation for using dual-stream in the introduction section.
3. It is recommended to conduct ablation studies on the impact of voxel grid size, as this can offer further insights.

**Suitability:**

2

---

### Official Review · Reviewer_VgGc · 2024-05-23

**Rating:** 4
**Confidence:** 3

**Summary:**

This paper proposes a new voxel-based framework for 3D point cloud object tracking. Authors claim that the previous widely-used point-based methods struggle to exploit 3D spatial information due to the permutation invariance pooling and set abstraction operations. Therefore, a new method named VoxelTrack is proposed to model 3D spatial information precisely. The VoxelTrack utilizes a voxel-based backbone and can consume current and previous frames using a Dual-Stream Voxel Representation Learning module. Finally, the Box Regression module can predict the tracklet by utilizing the output of the cross-iterative feature fusion module. The extensive experimental results demonstrate the effectiveness of the proposed method. Better, the VoxelTrack achieves a 36FPS of inference efficiency when using a single TITAN RTX GPU.

**Strengths:**

1. The proposed VoxelTrack is computing saving, which have a strong potential make a deployment on the vehicle or other real-world applications.
2. The paper is well-organized, and the proposed framework is technically sound.
3. The extensive experimental results demonstrate the effectiveness of the proposed VoxelTrack.

**Limitations:**

First, the authors should explain why the point-based algorithms cannot capture 3D spatial information well. To my knowledge, PointNet also adopted a concatenation operation of the permutation invariance pooled feature with the **original points**, which can capture both geometric and spatial information. The second claim, *the adopted set abstraction operation hardly handles density-inconsistent point clouds, also prevents 3D spatial information from being modeled*, is vague. The authors should provide more explanation to make it clean. Therefore, the motivation of authors using the voxel-based method to substitute the point-based method is not solid.

Second, as shown in Fig 4, The large and small voxel branch consumes two tensors in $\mathbb{R}^{W\times L\times H\times C}$ and $\mathbb{R}^{2W\times 2L\times 2H\times C}$ respectively, and output two feature maps with identical dimensions. However, the cascade pipeline of the Dual-stream Voxel Representation Learning module has a feature pyramid that down-samples the feature map at each level. These two figures state inconsistently.

Third, VoxelTrack is not the first paper to use a voxel-based method for object detection or tracking (or both), which limits the novelty of the proposed method.

**Suitability:**

3

---

### Official Review · Reviewer_rKUa · 2024-06-02

**Rating:** 3
**Confidence:** 3

**Summary:**

In this paper, a novel framework called VoxelTrack is introduced. VoxelTrack voxelizes point clouds into 3D voxels and extracts their features using sparse convolution blocks, effectively capturing precise 3D spatial information for accurate position prediction of tracked objects. Additionally, VoxelTrack employs a dual-stream encoder with a cross-iterative feature fusion module to enhance fine-grained 3D spatial information modeling. This approach simplifies the tracking pipeline with a single regression loss.

**Strengths:**

1. It simplifies the previous method by using Voxel Representation.
2. "CIF” (cross-iterative feature fusion module) helps to aggregate multi-scale features.

**Limitations:**

1. Are there any similar designs of CIF in point-based methods? Please summarize it in related works and compare them in experiments.
2. I think "Dual-Stream Voxel Representation Learning" is time-consuming since you need to extract features twice. With this consideration, how to extend your framework to perform the multi-object tracking, which is more useful in practices.

**Suitability:**

2

---

### Meta-Review · Area_Chair_znkw · 2024-07-02

**Recommendation:** Accept (Poster)
**Confidence:** 4

**Metareview:**

Three reviewers have reviewed the paper. The advantages include: The proposed method simplifies previous approaches using Voxel Representation; The Cross-Iterative Feature Fusion (CIF) module aggregates multi-scale features, enhancing effectiveness; VoxelTrack is computationally efficient, making it suitable for real-world applications like vehicle deployment; The paper is well-organized, technically sound, and easy to understand; Extensive experimental results support the effectiveness of VoxelTrack and its superiority over point-based backbones; The dual-stream encoder enhances multi-scale information, boosting overall performance. The disadvantages include the following: The authors need to provide a more solid justification for using voxel-based methods over point-based algorithms; The authors should clarify the claim that the set abstraction operation struggles with density-inconsistent point clouds.

Overall, the paper has a good quality and can be accepted to the conference. All reviewers support accepting this paper. The ACs do not find significant reasons to overturn reviewers' comments and thus recommend accepting this paper.